# How Do Large Language Models Capture the Ever-changing World Knowledge? A Review of Recent Advances

**Zihan Zhang[1]\*, Meng Fang[2]\*, Ling Chen[1], Mohammad-Reza Namazi-Rad[3], Jun Wang[4]**

[1]University of Technology Sydney  [2]University of Liverpool
[3]University of Wollongong  [4]University College London
Zihan.Zhang-5@student.uts.edu.au, Meng.Fang@liverpool.ac.uk
Ling.Chen@uts.edu.au, mrad@uow.edu.au, junwang@cs.ucl.ac.uk

## Abstract

Although large language models (LLMs) are impressive in solving various tasks, they can quickly be outdated after deployment. Maintaining their up-to-date status is a pressing concern in the current era. This paper provides a comprehensive review of recent advances in aligning LLMs with the ever-changing world knowledge without re-training from scratch. We categorize research works systemically and provide in-depth comparisons and discussion. We also discuss existing challenges and highlight future directions to facilitate research in this field [1].

## 1 Introduction

Large language models (LLMs) (Brown et al., 2020; Ouyang et al., 2022; Chowdhery et al., 2022; Zhang et al., 2022; OpenAI, 2023b; Touvron et al., 2023; Anil et al., 2023) trained on massive corpora from various sources (*e.g.*, Wikipedia, Books, Github) implicitly store enormous amounts of world knowledge in their parameters (Petroni et al., 2019; Roberts et al., 2020; Jiang et al., 2020), enabling them to act as versatile foundation models for performing various natural language processing (NLP) tasks directly through in-context learning (Liu et al., 2023b; OpenAI, 2023b; Bubeck et al., 2023; Kamalloo et al., 2023) or for further fine-tuning for domain-specific uses (Singhal et al., 2022; Google, 2023; Liu and Low, 2023).

Despite their impressive performance, LLMs are static after deployment, and there is no mechanism to update themselves or adapt to a changing environment (Kasai et al., 2022; Bubeck et al., 2023). Our world, however, is dynamic and constantly evolving. As shown in Fig.1, the static nature of trained LLMs makes the memorized knowledge

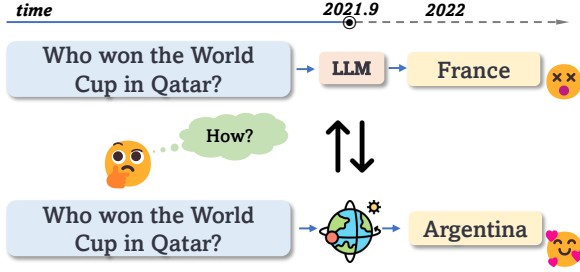

Figure 1: A trained LLM is static and can be outdated (*e.g.*, ChatGPT; OpenAI 2022). How can LLMs be aligned to the ever-changing world knowledge efficiently and effectively?

quickly obsolete, which often causes hallucinations, rendering them unreliable for knowledge-intensive tasks (Lazaridou et al., 2022; Luu et al., 2022; Ji et al., 2023; Si et al., 2023). In the era of LLMs, ensuring their alignment with the ever-changing world knowledge and maintaining their up-to-date status after deployment is a pressing concern because many users and downstream applications rely on them. Unfortunately, simply re-training LLMs with the latest information is infeasible due to prohibitive costs (Patterson et al., 2021).

Intuitively, to update an LLM, one can either replace the obsolete knowledge stored *implicitly* in the model with new ones by modifying its parameters, or override the outdated model outputs using new information *explicitly* retrieved from the world. Tremendous work has been proposed in the literature to implicitly or explicitly refresh deployed LLMs; however, these approaches, scattered among various tasks, have not been systematically reviewed and analyzed.

In this review, we survey the recent compelling advances in aligning deployed LLMs with the ever-changing world knowledge. We categorize research works systemically and highlight representative approaches in each category (§2) and provide in-depth comparison with discussion for insights (§3). Lastly, we discuss potential future directions to

---

\*Equal contribution

[1]We release the paper list at https://github.com/hyintell/awesome-refreshing-llms and will periodically update it.

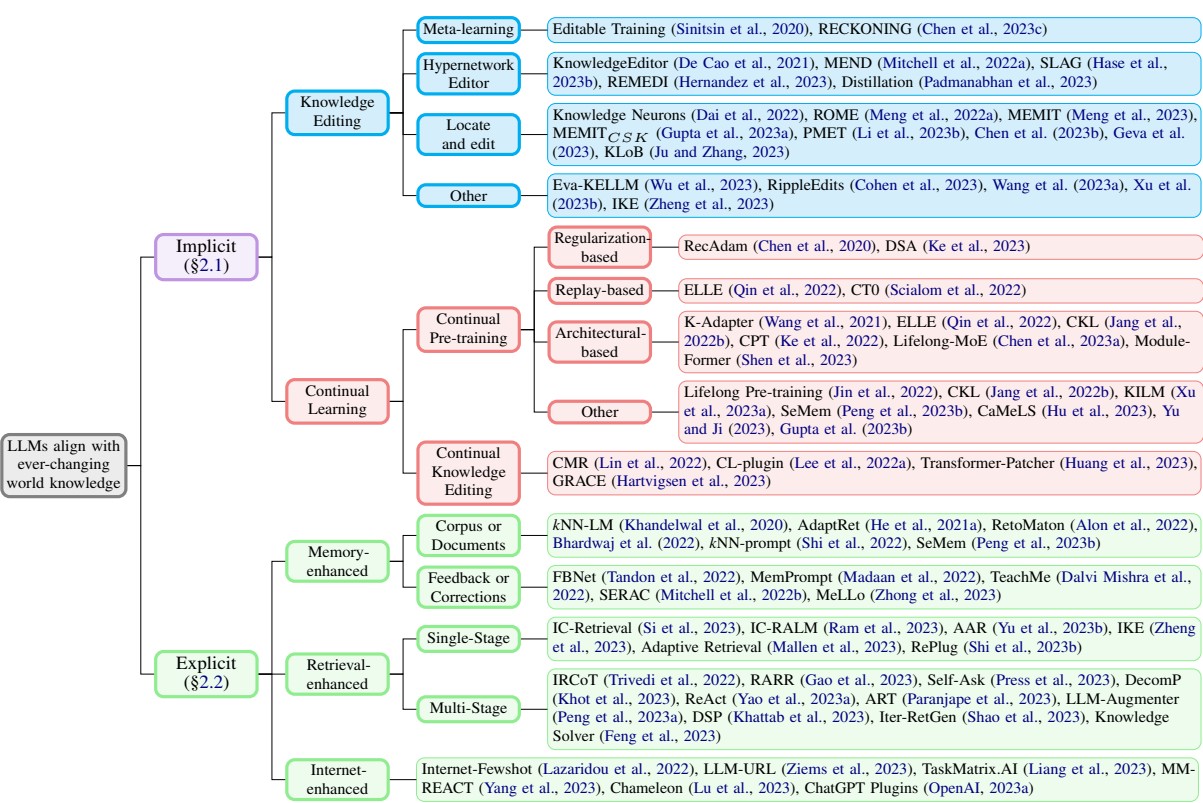

Figure 2: Taxonomy of methods to align LLMs with the ever-changing world knowledge (due to space limitation, please refer to Appendix A.2 for a complete review). **Implicit** means the approaches seek to directly alter the knowledge stored in LLMs (*e.g.*, parameters) (§2.1), while **Explicit** means more often incorporating external resources to override internal knowledge (*e.g.*, search engine) (§2.2).

facilitate research in this field (§4).

**Comparison with Related Work** To the best of our knowledge, surveys on this topic are scarce. Closest to our work, AlKhamissi et al. (2022) review pre-trained language models (LMs) as knowledge bases (KBs) and review a set of aspects that a LM should have to fully act as a KB; Cao et al. (2023) further divide the life cycle of knowledge in LLMs into five periods and survey how knowledge circulates; Yao et al. (2023b) conduct an empirical analysis of existing knowledge editing methods. Despite partially overlapping with our discussion of knowledge editing in §2.1.1, they only touch a subset of the scope that our survey studies and ignore other potentials in aligning LLMs with the world knowledge. Mialon et al. (2023); Wang et al. (2023b); Qin et al. (2023b) study augmented, interactive, and tool learning of LLMs respectively, which share different goals from ours. Previous knowledge-enhanced LMs surveys (Zhu et al., 2021; Wei et al., 2021; Yu et al., 2022; Yin et al., 2022; Zhen et al., 2022) focus on injecting knowledge into LMs, typically requiring modify-

ing the model's architecture or re-training. Instead, we focus on the potential of how deployed LLMs capture the *ever-changing* world knowledge effectively and efficiently without re-training. Wang et al. (2023c) provide a comprehensive review of forgetting in deep learning that is not limited to continual learning. Pan et al. (2023) review potential approaches that unify knowledge graphs (KGs) and LLMs. While structural knowledge, such as KGs, can broadly be categorised as explicit knowledge and augmented to LLMs for new knowledge, KG is static after creation and can still be outdated (Ji et al., 2022). New information or discoveries not yet incorporated into KGs may lead to outdated knowledge. However, how to efficiently update KGs is out of the scope of this survey.

## 2 Taxonomy of Methods

Based on whether the method tends to directly alter the knowledge stored implicitly in LLMs, or leverage external resources to override the outdated knowledge, we roughly categorize them as *implicit* (§2.1) or *explicit* (§2.2) approaches. Fig.2 provides a summary of representative works from each cate-

gory (See Fig.6 in Appendix for a complete review). Detailed descriptions of the methods can be found in Appendix A.1.

## 2.1 Implicitly Align LLMs with World Knowledge

Previous studies have shown that LLMs can implicitly memorize knowledge in their large number of parameters after being pre-trained on massive corpora (Petroni et al., 2019; Roberts et al., 2020; Jiang et al., 2020; Singhal et al., 2022). To keep LLMs up-to-date and align with the current world knowledge, the straightforward way is to alter the model's behaviour from *themselves* to generate desired outputs. Naively, one can regularly *re-train* the model from scratch or *fine-tune* the model with the latest corpora to align with current world knowledge. However, re-training is expensive and environmentally unfriendly (Patterson et al., 2021), especially in the era of LLMs with billions of parameters. Fine-tuning without constraints may have a "butterfly effect" and affect other knowledge or skills present in the model (Kirkpatrick et al., 2017; Li et al., 2022; AlKhamissi et al., 2022). To cope with this issue, this line of work aims to design better strategies to modify the internal states of LLMs in a more controllable and efficient way, which can be categorized into *knowledge editing* (§2.1.1) and *continual learning* (§2.1.2).

### 2.1.1 Knowledge Editing

Since tuning LLMs to learn new knowledge can be prohibitively expensive (Patterson et al., 2021), researchers seek efficient methods to directly update more specific, localized, or fine-grained knowledge that is preserved in LLMs (Mitchell et al., 2022a). Knowledge editing (KE) is an arising and promising research area that aims to alter the parameters of some specific knowledge stored in pre-trained models so that the model can make new predictions on those revised instances while keeping other irrelevant knowledge unchanged (Sinitsin et al., 2020; De Cao et al., 2021; Mitchell et al., 2022a; Meng et al., 2022a; Hase et al., 2023b; Meng et al., 2023). In this section, we categorize existing methods into *meta-learning*, *hypernetwork*, and *locate-and-edit*-based methods.

**Meta-learning.** This line of work generally focuses on the *intrinsic* editability of the model itself, aiming to modify the model parameters so that they can be easily updated during inference (De Cao

et al., 2021; Mitchell et al., 2022a). Sinitsin et al. (2020) propose a model-agnostic meta-learning-based (Finn et al., 2017) method that trains neural networks in a way that the trained parameters can be easily edited afterwards. Chen et al. (2023c) introduce a two-loop framework. In the inner training loop, they employ a few gradient updates to enable a pre-trained GPT-2 model (Radford et al., 2019) to efficiently memorize external knowledge. Subsequently, in the outer loop, the model parameters are dynamically adjusted through optimal meta-parameter learning to incorporate additional knowledge that aids reasoning tasks.

**Hypernetwork Editor.** In contrast to pre-modifying the pre-trained language model, an alternative approach in the field involves training *extrinsic* editors that update knowledge during test time, thereby avoiding any modifications to the base model. De Cao et al. (2021) reframe editing the knowledge of a model as a *learning-to-update* problem. Specifically, given a single data instance that needs to be updated, their trained hypernetwork (Ha et al., 2017) predicts a shift $\Delta\theta$ such that $\theta' = \theta + \Delta\theta$, where $\theta$ is the original pre-trained LM weights and $\theta'$ is the updated weights. To keep editing effective while being easy to scale to larger LMs with billions of parameters, Mitchell et al. (2022a) decompose weight updates into low-rank components (Hu et al., 2022), thus making it possible to scale to LLMs. Orthogonal to Mitchell et al. (2022a), Hase et al. (2023b) introduce a new training objective considering sequential, local, and generalizing model updates. Although scaled beyond a single edit, their edit success rate significantly degrades when performing larger edits simultaneously. Unlike the above methods that operate on the model's weight, Hernandez et al. (2023) perform edits on the representation level. Padmanabhan et al. (2023) employ knowledge distillation to transfer knowledge generated from a teacher model to a student model.

**Locate and Edit.** Generally, this line of work adopts the *locate and edit* pattern: they first identify the location of specific knowledge stored in the model via different assumptions, then directly modify the weights or representations to update knowledge. Inspired by the findings that feed-forward networks (FFN) in Transformer (Vaswani et al., 2017) are key-value memories (Geva et al., 2021), Dai et al. (2022) introduce the *knowledge neurons*

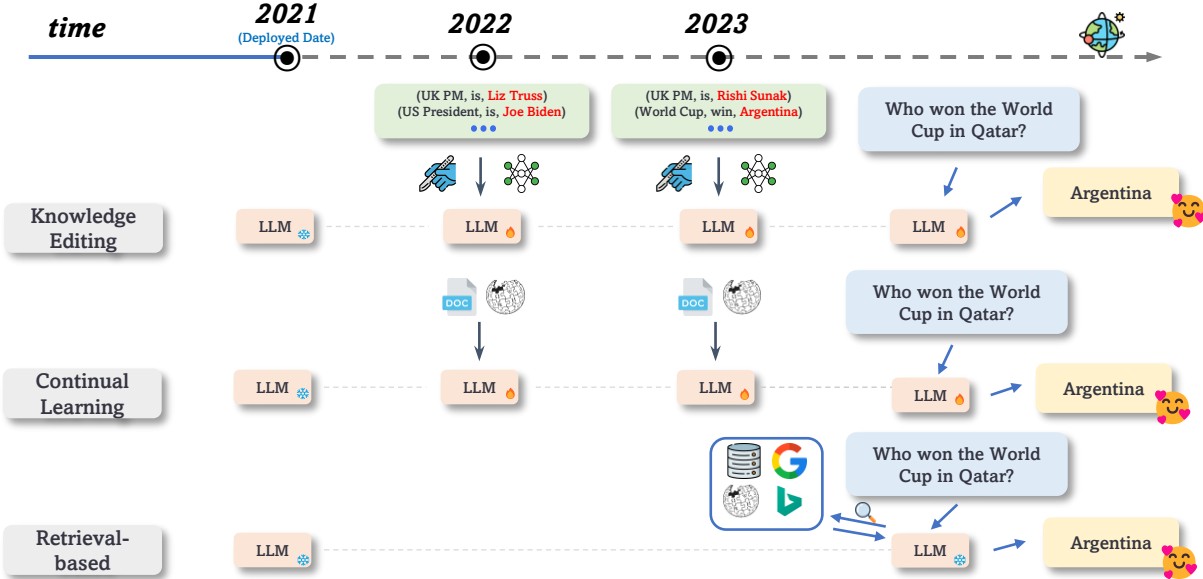

Figure 3: A high-level comparison of different approaches.

concept and propose a gradient-based knowledge attribution method to identify these knowledge neurons in FFNs. Further, without fine-tuning, they directly modify the corresponding value slots (*e.g.*, embeddings) in the located knowledge neurons and successfully update or delete knowledge, demonstrating a preliminary potential to edit knowledge in LMs.

Different from Geva et al. (2021)'s per-neuron view, Meng et al. (2022a) conduct casual tracing analysis on GPT-2 and hypothesize that the Transformer MLP can be viewed as a linear associative memory. They verify their hypothesis by directly updating the middle-layer MLP weights with a rank-one update (Bau et al., 2020). Following Meng et al. (2022a)'s work, Meng et al. (2023) propose a scalable multi-layer method to update an LLM with thousands of facts simultaneously, significantly improves editing efficiency while maintaining generalization and specificity. Gupta et al. (2023a) further adapt it to fix commonsense mistakes. Li et al. (2023b) find that Multi-Head Self-Attention (MHSA) weights do not require updating when introducing new knowledge. Based on this, they propose precisely updating FFN weights by simultaneously optimizing the Transformer component hidden states of MHSA and FFN to memorize target knowledge. Chen et al. (2023b) propose an architecture-adapted multilingual integrated gradients method to localize knowledge neurons precisely across multiple architectures and languages. Geva et al. (2023) analyze the internal recall pro-

cess of factual associations in auto-regressive LMs, opening new research directions for knowledge localization and model editing.

**Other.** Wu et al. (2023) propose an evaluation framework and dataset for measuring the effectiveness of knowledge editing of LLMs, as well as the ability to reason with the altered knowledge and cross-lingual knowledge transfer. Similarly, Cohen et al. (2023) evaluate the implications of an edit on related facts and show that existing methods fail to introduce consistent changes in the model's knowledge. Ju and Zhang (2023) propose an evaluation benchmark for locate-and-edit-based methods, aiming to reassess the validity of the locality hypothesis of factual knowledge. Wang et al. (2023a) and Xu et al. (2023b) take multilingual into account and extend existing knowledge editing methods into cross-lingual scenarios.

### 2.1.2 Continual Learning

Continual learning (CL) aims to enable a model to learn from a continuous data stream across time while reducing catastrophic forgetting of previously acquired knowledge (Biesialska et al., 2020). With CL, a deployed LLM has the potential to adapt to the changing world without costly re-training from scratch (Bubeck et al., 2023). In this section, we introduce approaches that employ CL for aligning LLMs with the current world knowledge, including *continual pre-training* and *continual knowledge editing*.

**Continual Pre-training.** Unlike traditional continual learning, which sequentially fine-tunes a pre-trained LM on some specific downstream tasks (*e.g.*, QA, text classification), *continual pre-training* is used to further pre-train an LM to acquire new knowledge, where the data corpus is usually *unsupervised* (Gururangan et al., 2020; Ke and Liu, 2023). Since our target is the versatile foundation LLMs (*e.g.*, GPT-4) that can be applied to many different use cases rather than a fine-tuned model designed for a specific task, we focus on the literature on continual pre-training.

Early works (Gururangan et al., 2020; Röttger and Pierrehumbert, 2021; Lazaridou et al., 2021; Dhingra et al., 2022) empirically analyze continuing LM pre-training on emerging domain or temporal data, showing the potential to update the base LM with new knowledge. Jang et al. (2022b) explicitly categorize world knowledge as time-invariant, outdated, and new knowledge, which should be retained, acquired, and updated respectively by an LM when learning continually. Jin et al. (2022); Jang et al. (2022a,b) additionally implement traditional CL methods to alleviate *catastrophic forgetting*, a phenomenon in which previously learned knowledge or abilities are degraded due to overwritten parameters (Kirkpatrick et al., 2017). Among the literature, CL methods can be mainly categorized into ① **Regularization**, ② **Replay**, and ③ **Architectural** -based methods.

① **Regularization**. To mitigate forgetting, regularization-based methods apply regulations to penalize the changes of the critical parameters learned from previous data. Chen et al. (2020) improve the traditional EWC (Kirkpatrick et al., 2017) by recalling previously learned knowledge through the pre-trained parameters, and the method continually learns new information using a multi-task learning objective. Ke et al. (2023) compute the importance of each unit (*i.e.*, attention head and neuron) to the general knowledge in the LM using a proxy based on model robustness to preserve learned knowledge. When continually learning new domains, the approach prevents catastrophic forgetting of the general and domain knowledge and encourages knowledge transfer via soft-masking and contrastive loss.

② **Replay**. These methods generally reduce forgetting by replaying previous training data when learning new data. Assuming that the initial pre-training corpus is available, He et al. (2021b) use a gradual decay mix-ratio to adjust the quantity of the pre-training corpus mixed in the new data when learning sequentially. ELLE (Qin et al., 2022) and CT0 (Scialom et al., 2022) also mix the old data while learning new data. However, ELLE starts the pre-training from a newly initialized and relatively small BERT (Devlin et al., 2019) and GPT (Radford et al., 2018), while CT0 continues learning from T0-3B (Sanh et al., 2022), a pre-trained and instruction-tuned model.

③ **Architectural**. These methods typically alleviate forgetting by using different subsets of parameters for distinct tasks or domains. Wang et al. (2021); Hu et al. (2022); Ke et al. (2022) freeze the original parameters of the LM to preserve the learned knowledge and add lightweight tunable parameters for continually learning new knowledge. Wang et al. (2021) add separate adapters (Houlsby et al., 2019) for each new task, while Ke et al. (2022) let all domains share adapters and employ task masks to protect critical neurons from being updated. DEMix-DAPT (Gururangan et al., 2022) replaces every FFN layer in Transformer with a separate domain expert mixture layer, containing one expert per domain. When learning new knowledge, they only train the newly added expert in each DEMix layer while fixing all other experts. Similarly, Lifelong-MoE (Chen et al., 2023a) progressively expands experts to increase model capacity for learning new knowledge, and mitigates forgetting by freezing previously trained experts and gatings with output-level regularization. Qin et al. (2022) enlarge the model's width and depth to attain learning efficiency and employ memory replay to reduce forgetting.

④ **Other Methods**. Hu et al. (2023) meta-trains an importance-weighting model to reweight the per-token loss of the continual data stream, intending to quickly adapt the base LM to new knowledge. Peng et al. (2023b) apply *k*NN-LM (Khandelwal et al., 2020) to continual learning from streaming data and selectively store hard cases in a non-parametric memory, significantly improving the data-wise and model-wise scalability. Yu and Ji (2023) assess self-information-update in LLMs via CL and mitigate exposure bias by incorporating the selection of relevant facts into training losses.

**Continual Knowledge Editing.** Lin et al. (2022); Lee et al. (2022a); Huang et al. (2023) and Hartvigsen et al. (2023) propose a more realistic setting that a deployed LM should be constantly

| Category | Representative Method | Base LM | LM Params | Augmentation | No Training | Black-box |
|---|---|---|---|---|---|---|
| **Knowledge Editing** | MEND (Mitchell et al., 2022a) | T5 (11B) | ❄ | auxiliary model | ✗ | ✗ |
| | ROME (Meng et al., 2022a) | GPT-J (6B) | 🔥 | – | ✔ | ✗ |
| | CaliNET (Dong et al., 2022) | T5 (0.7B) | ❄ | +params | ✗ | ✗ |
| | MEMIT (Meng et al., 2023) | GPT-NeoX (20B) | 🔥 | – | ✔ | ✗ |
| **Continual Learning** | K-Adapter (Wang et al., 2021) | RoBERTa (0.3B) | ❄ | +params | ✗ | ✗ |
| | CT0 (Scialom et al., 2022) | T0 (3B) | 🔥 | memory | ✗ | ✗ |
| | DSA (Ke et al., 2023) | RoBERTa (0.1B) | 🔥 | – | ✗ | ✗ |
| **Memory-enhanced** | MemPrompt (Madaan et al., 2022) | GPT-3 (175B) | ❄ | memory+retriever | ✔ | ✔ |
| | SERAC (Mitchell et al., 2022b) | T5 (0.7B) | ❄ | memory +auxiliary model | ✗ | ✔ |
| | MeLLo (Zhong et al., 2023) | GPT-3.5 (175B) | ❄ | memory+retriever | ✔ | ✔ |
| **Retrieval-enhanced** | IRCoT (Trivedi et al., 2022) | GPT-3.5 (175B) | ❄ | retriever | ✔ | ✔ |
| | RARR (Gao et al., 2023) | PaLM (540B) | ❄ | search engine +auxiliary model | ✔ | ✔ |
| | DecomP (Khot et al., 2023) | GPT-3 (175B) | ❄ | retriever | ✔ | ✔ |
| | ReAct (Yao et al., 2023a) | PaLM (540B) | ❄ | search engine | ✔ | ✔ |
| | FLARE (Jiang et al., 2023) | GPT-3.5 (175B) | ❄ | retriever/search engine | ✔ | ✔ |
| **Internet-enhanced** | Lazaridou et al. (2022) | Gopher (280B) | ❄ | search engine | ✔ | ✔ |
| | CRITIC (Gou et al., 2023) | GPT-3.5 (175B) | ❄ | various tools | ✔ | ✔ |
| | Chameleon (Lu et al., 2023) | GPT-4 (?B) | ❄ | various tools | ✔ | ✔ |

Table 1: Comparison between representative methods (refer to Appendix A.2 for a complete review). 🔥 means the parameters of the original LM are modified, while ❄ means they are unchanged; **Augmentation** means additional components used; **No Training** indicates the method does not require additional training; **Black-box** refers to whether the method suits non-publicly available models (*e.g.*, no model architecture, parameters, activations, or gradients are available).

corrected to fix its prediction errors, showing the potential to align the model with the latest world knowledge. Lin et al. (2022) benchmark the continual model refinement problem by implementing traditional CL methods. Lee et al. (2022a) and Hartvigsen et al. (2023) freeze the LM's original parameters and continually introduce trainable neurons to the FFN layer to rectify problematic model behaviors. In contrast, Hartvigsen et al. (2023) learn to cache a chosen layer's activations in a key-value-based codebook and retrieve activations when previous similar edits have been performed. Without influencing unrelated inputs, it can efficiently edit the model thousands of times in a row while generalizing edits to previously unseen inputs.

## 2.2 Explicitly Align LLMs with World Knowledge

Although altering the knowledge implicitly stored in LLMs has shown to be effective (Jang et al., 2022b; Meng et al., 2023), it remains unclear whether it will affect the models' general abilities due to the complexity of neural networks. In contrast, explicitly augmenting LLMs with the latest information retrieved from various sources can effectively adapt the models to new world knowledge without affecting the original LLMs (Mialon et al., 2023). However, previous retrieval-augmented

methods (Karpukhin et al., 2020; Guu et al., 2020; Lewis et al., 2020; Izacard et al., 2022; Borgeaud et al., 2022; Jiang et al., 2022; Kaur et al., 2022) usually jointly train a retriever and an LM in an end-to-end fashion, making it challenging to apply to a deployed LLM (*e.g.*, GPT-3). Recently, researchers have focused on equipping a fixed LLM with external memory (*memory-enhanced*; §2.2.1), an off-the-shelf retriever (*retrieval-enhanced*; §2.2.2), or Internet (*Internet-enhanced*; §2.2.3) to cope with this issue.

### 2.2.1 Memory-enhanced Methods

Pairing a static LLM with a growing non-parametric memory enables it to capture information beyond its memorized knowledge during inference (Wu et al., 2022). The external memory can store a recent *corpus* or *feedback* that contains new information to guide the model generation.

**Storing Corpus or Documents.** *k*NN-LM (Khandelwal et al., 2020) stores every <context, token> as key-value pairs from a corpus in memory. During inference, it calculates the probability of the next token by interpolating a fixed LM with a distribution retrieved from the *k* nearest tokens in the memory. Following this vein, He et al. (2021a); Drozdov et al. (2022); Alon et al. (2022) improve the efficiency of *k*NN-LM by skipping

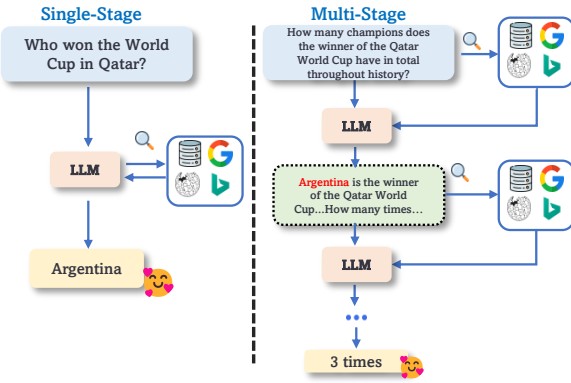

Figure 4: **Single-Stage** (left) typically retrieves once, while **Multi-Stage** (right) involves multiple retrievals or revisions to solve complex questions (§2.2.2).

unnecessary retrieval. Meng et al. (2022b) build an additional graph neural network to aggregate information from the retrieved context for better generation. Peng et al. (2023b) improve the scalability of *k*NN-LM for continual learning, while Shi et al. (2022) apply it for zero-shot inference on downstream tasks.

**Storing Feedback or Corrections.** Inspired by the fact that humans can learn from past mistakes, this line of work stores user feedback in memory to fix the model's problematic predictions and avoids similar errors in the future. By querying the memory, the base LLM gains *editability* to update its outdated knowledge. Kassner et al. (2021); Tandon et al. (2022) train an auxiliary corrector to apply feedback to repair the model output. Dalvi Mishra et al. (2022) allow users to interact with the system to check its facts and reasoning and correct it when it is wrong. Similarly, Madaan et al. (2022) equip GPT-3 with a growing memory, where the key is a misunderstanding question, and the value is the corrective feedback. Instead of storing user feedback, Mitchell et al. (2022b); Zhong et al. (2023) explicitly preserve updated knowledge in memory. Given an input, Mitchell et al. (2022b) first apply a classifier to determine if a relevant edit exists in the memory and perform knowledge updating through a counterfactual model. Conversely, Zhong et al. (2023) decompose complex questions and ask the base model to generate a temporary answer. They revise the model output when the generated answer contradicts the retrieved facts from memory.

### 2.2.2 Retrieval-enhanced Methods

Leveraging an off-the-shelf retriever and the in-context learning ability of LLMs (Brown et al.,

2020), this line of work designs better retrieval strategies to incorporate world knowledge into a fixed LLM through prompting, which can be divided into *single-stage* and *multi-stage* (Fig.4).

**Single-Stage.** To ground the model with external knowledge during generation, Ram et al. (2023); Si et al. (2023) adopt zero-shot and few-shot retrieval respectively and directly prepend the retrieved documents to the input without changing the base LLM. Zheng et al. (2023) retrieve similar edit demonstrations for each input and perform in-context knowledge editing. Compared with gradient-based knowledge editing (§2.1.1), they have competitive editing performance with fewer side effects. Arguing that the general-purpose retrievers could be sub-optimal, Yu et al. (2023b) adopt a small source LM to provide LM-preferred signals to train an adaptive retriever. Mallen et al. (2023) employ a heuristic based on entity popularity and only retrieve relevant context when the input questions are less popular, which improves performance and reduces inference costs. Unlike above, to address the limited model's context length, Shi et al. (2023b) prepend each retrieved document separately to an LLM and then ensemble output probabilities from different passes.

**Multi-Stage.** When solving complex questions, retrieving information only once based on the input is often inadequate. This branch of work aims to transform single-stage retrieval into multi-stage retrieval in order to solve complex tasks, usually by leveraging reasoning. Trivedi et al. (2022) interleave knowledge retrieval with chain-of-thoughts (CoT; Wei et al. 2022) generation to solve complex multi-step reasoning questions. Similarly, Press et al. (2023); Khot et al. (2023); Yao et al. (2023a); Jiang et al. (2023); Shao et al. (2023) decompose questions into sub-questions to provide a specific context for retrieval with model generation. Paranjape et al. (2023); Chen et al. (2023d); Inaba et al. (2023) further enable the usage of different tools to solve various tasks. Unlike the simple *retrieve-then-read* paradigm, Khattab et al. (2023) pass intermediate messages between an LLM and a retriever; Gao et al. (2023); He et al. (2022); Zhao et al. (2023); Yu et al. (2023a) retrieve after generation and perform post-edit revisions for more faithful outputs. Peng et al. (2023a) iteratively revise ChatGPT to improve model responses using feedback and external knowledge. Feng et al.

(2023) teach LLMs themselves to search for knowledge from external knowledge graphs (KGs) via prompting and simplify searching into a multi-hop decision sequence, allowing explainable decision-making of the processes.

### 2.2.3 Internet-enhanced Methods

Prior retrieval-augmented work relies on *static* or *offline* knowledge sources (*e.g.*, Wikipedia dump), which may not be sufficiently up-to-date or complete for tasks that require the latest knowledge (Kasai et al., 2022; Zhang et al., 2023; Li et al., 2023a). A recent trend uses the whole web as the knowledge source and equips LLMs with the Internet to support real-time information seeking (Nakano et al., 2022; Menick et al., 2022; Komeili et al., 2022; Shuster et al., 2022; Qin et al., 2023a; Liu et al., 2023a). Lazaridou et al. (2022) augment few-shot QA prompting with the context retrieved from Google search. Press et al. (2023); Jiang et al. (2023) interleave reasoning with web search. Recently, tools such as LangChain (Chase, 2022) and ChatGPT Plugins (OpenAI, 2023a) connect a deployed LLM to the Internet without training, making them more powerful for solving knowledge-intensive tasks. Beyond search engines, Yao et al. (2023a); Liang et al. (2023); Paranjape et al. (2023); Yang et al. (2023); Gou et al. (2023); Lu et al. (2023) treat LLMs as central planners and compose various plug-and-play tools for solving complex questions.

## 3 Comparison and Discussion

We present the comparison of different methods in Table 1 and in Fig.3, and the characteristics of different methods in Table 2 in Appendix.

**Discussion of Implicit Methods (§2.1).** Compared to naive re-training or fine-tuning, KE and CL can effectively update obsolete knowledge in LLMs while minimizing interference on irrelevant ones. We identify their major differences: ① **Scale**. Existing KE methods focus on updating small-scale and localized knowledge, typically on synthetic fact pairs (Mitchell et al., 2022a; Meng et al., 2022a). While one can perform thousands of edits simultaneously (Meng et al., 2023), updating enormous knowledge in LLMs may be cumbersome. In contrast, CL enhances models' adaptability via tuning larger-scale parameters, thus updating more knowledge at scale (Jang et al., 2022b). However, KE provides fine-grained controllability

when specific knowledge needs to be altered, which is unachievable by CL; ② **Forgetting**. Applying KE methods on LLMs frequently in response to the ever-changing world is sub-optimal due to catastrophic forgetting (Huang et al., 2023; Hartvigsen et al., 2023); CL mitigates this issue when learning new knowledge; ③ **Cost**. CL is generally more computationally expensive than KE due to larger-scale weight updating.

**Discussion of Explicit Methods (§2.2).** Explicit methods use new knowledge retrieved from the world to override old knowledge in an LLM during generation. Despite being effective, memory- and retrieval-enhanced methods must periodically maintain the external memory and the knowledge sources in response to the ever-changing world (Kasai et al., 2022). Conversely, Internet-enhanced methods enable real-time knowledge seeking, although potentially suffering from noisy and low-quality web content (Li et al., 2023a; Luo et al., 2023). Compared to single-stage retrieval, multi-stage retrieval can solve more complex problems. Nevertheless, they may interrupt the generation with multiple retrievals or revisions, leading to considerable inference overheads (Shao et al., 2023).

**Updating LLMs Implicitly or Explicitly?** We observe an increasing trend of explicitly aligning LLMs with world knowledge while keeping the model untouched (Table 3 in Appendix). Compared to explicit approaches: ① **Applicability**. Implicit methods usually require modifying LLM's parameters or gradients, making it challenging to update closed-source models; ② **Side Effects**. Although constraints have been added to avoid editing irrelevant knowledge (Mitchell et al., 2022a; Meng et al., 2023) or forgetting general knowledge (Jang et al., 2022b), modifying the LLM's parameters inevitably has side effects that may hurt the performance, which is hard to estimate due to the complexity of neural networks (Brown et al., 2023); ③ **Efficiency**. Implicit methods typically require training, while most explicit methods leverage a fixed LLM and an off-the-shelf retriever, erasing the necessity of training. However, explicit methods do not directly modify the intrinsic knowledge within LLMs, but instead rely on on-the-fly retrieval during inference, resulting in a notable increase in the computational cost of inference.

## 4    Challenges and Future Directions

**Robust and Efficient Knowledge Editing.**  KE offers fine-grained knowledge updating, which is desirable in some scenarios. Despite promising, KE is still in its infancy stage. ① **Various knowledge**. It is challenging to renew the internal knowledge stored in the parameters of LLMs, and existing efforts have only explored updating relational knowledge while overlooking other knowledge (Meng et al., 2023); ② **Edit dataset**. Current KE methods assume edited knowledge pairs exist, which must be annotated beforehand. In reality, how do LLMs know what knowledge is outdated and thus needs to be updated (Zhang and Choi, 2023; Yin et al., 2023)? ③ **Memorization mechanism**. Hase et al. (2023a) argue that the localization of specific knowledge via casual tracing may not be reliable, calling for a better understanding of the internal memorization of LLMs (Tirumala et al., 2022; Carlini et al., 2023); ④ **Generalization**. Recent studies (Onoe et al., 2023; Zhong et al., 2023) find that existing KE methods show little propagation of edited knowledge, meaning the LLM cannot make further reasoning based on the newly acquired knowledge; ⑤ **Effectiveness and efficiency**. Although early efforts have been made (Hernandez et al., 2023; Huang et al., 2023; Hartvigsen et al., 2023), methods to effectively, efficiently, and continually renew the knowledge of LLMs at scale have yet to be thoroughly explored.

**Efficient Continual Learning of LLMs.**  A continual pre-trained LLM can update its internal knowledge and adapt to the changing world, but maintaining the general knowledge required for downstream tasks without forgetting is challenging (Ke and Liu, 2023). Moreover, existing methods are limited to small-scale LMs, leaving CL of LLMs rarely studied. While parameter-efficient tuning (Ding et al., 2022) may be beneficial, it remains under-explored to align an LLM with the dynamic world via CL.

**Solving Knowledge Conflicts.**  Replacing old knowledge with new ones can cause knowledge conflicts regardless of using implicit or explicit methods. For implicit methods, these side effects are only evaluated in specific settings, and there is no idea of how the general skills of LLMs are impacted (Brown et al., 2023). For retrieval-based methods, knowledge retrieved from the world can contradict the knowledge memorized inside LLMs, and LLMs sometimes favour their internal knowledge rather than the provided context during generation (an example in Fig.5; Neeman et al. 2022; Li et al. 2022; Chen et al. 2022). While initial attempts have been made (Mallen et al., 2023; Zhou et al., 2023; Xie et al., 2023), they are still limited.

**Robust and Efficient Retrieval.**  Interacting with external resources can cause interruptions during generation, significantly increasing inference overheads, especially for multi-stage methods that involve multiple retrievals or revisions. Potential remedies may be efficient memory management (Peng et al., 2023b; Kang et al., 2023; Cheng et al., 2023) or selective retrieval that only consults external resources when necessary (Mallen et al., 2023). On the other hand, the retrieved context can be irrelevant and noisy, which may distract LLMs (Shi et al., 2023a; Luo et al., 2023), or too long, which exceeds the input limits and renders high cost (Shi et al., 2023b).

**Comprehensive Evaluation and Benchmarks.** Although approaches of different categories can align the trained LLMs with the changing world without re-training, their effectiveness is primarily evaluated on synthetic datasets in specific settings, which might not be comprehensive (Jang et al., 2022a,b; Hoelscher-Obermaier et al., 2023). Moreover, although efforts have been made to evaluate KE (Wu et al., 2023; Cohen et al., 2023; Ju and Zhang, 2023), there is no quantitative comparison of methods of different categories (*i.e.*, comparing KE vs. CL vs. retrieval-based methods), hindering their application in different scenarios. Lastly, existing benchmarks are too *static* to measure the dynamic world, which calls for real-time evaluation benchmarks (Liška et al., 2022; Kasai et al., 2022).

## 5    Conclusion

In this paper, we systematically review recent advances in aligning LLMs with the ever-changing world knowledge without re-training. We summarize existing approaches and categorize them based on whether they tend to directly alter the knowledge stored implicitly in LLMs, or leverage external resources to override the outdated knowledge. We comprehensively compare methods of different categories and point out challenges and future directions to facilitate research in this area.

## Limitations

In this paper, we systematically review recent advances in aligning LLMs with the ever-changing world knowledge without re-training. We compare our work with the related surveys in §1 and will periodically add related approaches. Despite our best efforts, there exist some limitations in this paper:

**Scope.** In this survey, we do not review knowledge-enhanced approaches that require retraining because we focus on the already trained (deployed) models and how to keep them up-to-date. We refer interested readers to the relevant knowledge-enhanced LMs surveys (Zhu et al., 2021; Wei et al., 2021; Yu et al., 2022; Yin et al., 2022; Zhen et al., 2022). Second, in terms of world knowledge, we focus on text-based knowledge and leave other kinds of knowledge, such as images, video, audio, etc., and structural knowledge, such as knowledge graphs (KGs) and databases, for future work. Third, we mainly review the cutting-edge approaches within three years (mostly in 2022 and 2023) in §2, mainly from the ACL, EMNLP, NAACL, TACL, NeurIPS, ICML, ICLR, arXiv. Despite our best efforts, by no means the surveyed methods are complete, and we may miss some important references. Lastly, we cannot afford all the technical details due to page limitations and may only provide brief introductions. We provide additional discussion of approaches in Appendix A.1.

**Taxonomy.** It should be noted that some approaches are hybrid and can be categorized into different branches. We mainly categorize them based on their main components or mechanism. For instance, all methods in §2.2 require retrieving from external resources. Memory-enhanced methods (§2.2.1) pay more attention to the design of external memory, while paying little attention to retrieval strategies.

**Empirical Comparison.** We provide detailed comparisons and discussions in §3 and potential future directions in §4. All the conclusions are proposed based on empirical summarization of existing works. However, as the field evolves fast, these empirical conclusions might be inapplicable. We will update the latest opinions timely. In addition, we do not provide quantitative comparisons through experiments since there is no unified evaluation benchmarks of different categories. Quan-

titative evaluation (benchmarks) is a challenging and interesting future direction to fairly compare methods of different categories to align LLMs with updated world knowledge (§4). We will leave quantitative comparisons and analysis as future work.

## Acknowledgements

This work is supported by TPG Telecom. We would like to thank anonymous reviewers for their valuable comments.

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

# A  Appendix

## A.1  Additional Description of Approaches

### A.1.1  Naive Approaches

Although more advanced approaches have been proposed, we introduce naive solutions for completeness in this section.

**Re-training.**  Intuitively, one can regularly re-train the model from scratch with the latest corpora to align with current world knowledge. However, this naive solution has clear downsides: (1) Re-training is both time and money expensive and environmentally unfriendly (Patterson et al., 2021), especially in the era of LLMs with billions of parameters. For instance, LLaMA-65B was trained for about one million GPU-hours and emitted more than a hundred tons of carbon (Touvron et al., 2023); (2) It is unrealistic to frequently re-training an LLM in response to the constantly changing world.

**Fine-tuning.**  Another simple approach is to periodically curate a small-scale dataset containing desired knowledge we wish the model to add, update, or delete, then fine-tune the model on the dataset. Despite being computationally cheaper than re-training, it still falls short in that, without constraints, directly fine-tuning the model may have a "butterfly effect" and affect other knowledge or skills present in the model (Li et al., 2022), causing degraded generalization (Mitchell et al., 2022a), catastrophic forgetting (Kirkpatrick et al., 2017; Zhu et al., 2020; AlKhamissi et al., 2022), or knowledge conflicts (Neeman et al., 2022).

**Constrained Fine-tuning.**  To solve part of above mentioned issues, Zhu et al. (2020) propose to only fine-tune the model on the small-scale modified facts set and add explicit constraints on the model weights so that the model learns to answer the modified facts while keeping the remaining knowledge intact. Specifically, they use various norms ($\mathcal{L}_0$, $\mathcal{L}_2$, and $\mathcal{L}_\infty$) to prevent the parameters of the fine-tuned model $\theta'$ from drifting too far from the original model parameters $\theta$. They further find that fine-tuning only the first and last layers of the Transformer model (Vaswani et al., 2017) results in better adaptation to the modified facts and better preservation of performance on the unmodified facts. However, the norm-based constraint on parameters ignores the highly non-linear nature of LMs and how parameters determine the outputs of the model, making their method potentially unreliable (De Cao et al., 2021). In addition, Mitchell et al. (2022a) confirm that constrained fine-tuning generally does not consistently provide edit generality.

Figure 5: An example of knowledge conflict of Chat-GPT (OpenAI, 2022). Even if the correct context is provided, ChatGPT still favours its internally memorized knowledge. The screenshot was taken in May 2023 for GPT-3.5 without web browsing.

### A.1.2 Knowledge Editing

To facilitate the development of this area, De Cao et al. (2021) formulate three desiderata that an ideal editing method should follow: ① **Generality**: the method should be capable of altering the knowledge of any LM that is not specifically trained to be editable (*e.g.*, PaLM, GPT-4, LLaMA); ② **Reliability**: the method should only update the targeted knowledge without influencing the rest of the knowledge in the LM. For instance, the answer to `"Who is the current Prime Minister of Australia?"` has changed from `"Scott Morrison"` to `"Anthony Albanese"` since 2022, updating the knowledge from `"Scott Morrison"` to `"Anthony Albanese"` should not change the knowledge `"Argentina won the 2022 World Cup"`; ③ **Consistency (Generalization)**: after updates, the model predictions should be consistent across semantically equivalent inputs (*e.g.*, correctly predicts `"Anthony Albanese"` to `"Who is the AU PM?"`). Beyond updating outdated knowledge, knowledge editing can also delete sensitive information for privacy issues or eliminate biases in the pre-training corpora.

However, not until recently, Onoe et al. (2023); Zhong et al. (2023) show that, after performing knowledge editing, the LLM does not really "learn" the updated knowledge and thus cannot *propagate* the new knowledge and make further inferences based on them. For instance, after learning that `"the current PM of Australia is Anthony Albanese"`, the model might not able to make predictions of `"Who is the spouse of the current PM of Australia?"`.

**Meta-learning.** Sinitsin et al. (2020), by constraining the training objective, encodes editability into the parameters of the model itself so that the model is "prepared" for incoming edits. While being effective and no new parameters are required, it does not conform to generality as it requires spe-

cialized training of the original model (De Cao et al., 2021). Moreover, to enforce the constraint that the editable model agrees with the original pretrained model's predictions, Sinitsin et al. (2020)'s method needs to retain a copy of the original model, which significantly consumes computation memory (Mitchell et al., 2022a). Chen et al. (2023c) also requires training of the original LM, which could be computationally expensive for larger LMs. In addition, whether it will influence other irrelevant knowledge in the model remains unknown, making the method potentially unreliable.

**Hypernetwork Editor.** De Cao et al. (2021) can be more efficient than Sinitsin et al. (2020), as it does not retain the copy of the original model nor compute higher-order gradients. However, it can only update a single fact rather than multiple facts in a row and fail to edit large models, leading to poor scalability (Mitchell et al., 2022a; Hase et al., 2023b). Mitchell et al. (2022a) improve De Cao et al. (2021)'s work and is stable to edit LMs from BERT-base (110M) (Devlin et al., 2019) to T5-XXL (11B) (Raffel et al., 2020). However, when editing multiple knowledge simultaneously, their edit success rate significantly degrades.

**Locate and Edit.** While simple, Dai et al. (2022) do not ensure reliability on other irrelevant knowledge and generalization on semantically equivalent inputs. Despite showing both generalization and specificity, Meng et al. (2022a) only edits a single fact at a time, making it impractical for large-scale knowledge updating in LLMs. Through casual tracing, Meng et al. (2023) identify and update the critical MLP layers in one go. However, Hase et al. (2023a) argue that the relation between localization and editing may be misleading as they can edit factual knowledge in different locations that are not suggested by casual tracing.

### A.1.3 Continual Learning

While knowledge editing provides a fine-grained control to update specific knowledge in LLMs, it often requires large amounts of supervised training data to make edits, which is non-trivial to create (Hartvigsen et al., 2023). In addition, when an LLM needs to quickly acquire new domain knowledge (e.g., legal or medical), such small-scale model edits may not be efficient. Moreover, after multiple parameter patches to a deployed model, its internal knowledge may conflict, leading to unpredictable behaviors (Mitchell et al., 2022a).

Sharing a related goal, continual learning (CL) aims to enable a model to learn from a continuous data stream across time while reducing catastrophic forgetting of previously acquired knowledge (Biesialska et al., 2020). In contrast to knowledge editing, CL generally updates models on a larger scale and works in long learning sequences with minimal memory overheads (Mitchell et al., 2022a). Hence, CL can also be used for deployed models to update their knowledge.

## A.2 The Complete Taxonomy of Methods

We list the complete taxonomy of methods to align LLMs with the ever-changing world knowledge in Fig.6 and the complete comparison of methods in Table 3. We also compare the characteristics of different methods in Table 2.

| Category | Large Scale | No Side Effects | Persistent |
|---|---|---|---|
| Knowledge Editing (§2.1.1) | ✗ | ✗ | ✔ |
| Continual Learning (§2.1.2) | ✔ | ✗ | ✔ |
| Retrieval-based (§2.2) | ✗ | ✔ | ✗ |

Table 2: High-level comparison of characteristics of different approaches.

| Category | Representative Method | Base LM | LM Params | Augmentation | No Training | Black-box |
|---|---|---|---|---|---|---|
| **Naive** | Re-training | – | 🔥 | – | ✗ | ✗ |
| | Fine-tuning | – | 🔥 | – | ✗ | ✗ |
| **Knowledge Editing** | De Cao et al. (2021) | BERT (0.1B) | ❄ | auxiliary model | ✗ | ✗ |
| | MEND (Mitchell et al., 2022a) | T5 (11B) | ❄ | auxiliary model | ✗ | ✗ |
| | SLAG (Hase et al., 2023b) | BERT (0.1B) | ❄ | auxiliary model | ✗ | ✗ |
| | RECKONING (Chen et al., 2023c) | GPT-2 (0.1B) | 🔥 | – | ✗ | ✗ |
| | ROME (Meng et al., 2022a) | GPT-J (6B) | 🔥 | – | ✔ | ✗ |
| | Knowledge Neurons (Dai et al., 2022) | BERT (0.1B) | 🔥 | – | ✔ | ✗ |
| | MEMIT (Meng et al., 2023) | GPT-NeoX (20B) | 🔥 | – | ✔ | ✗ |
| | CaliNET (Dong et al., 2022) | T5 (0.7B) | ❄ | +params | ✗ | ✗ |
| | REMEDI (Hernandez et al., 2023) | GPT-J (6B) | 🔥 | auxiliary model | ✗ | ✗ |
| **Continual Learning** | DSA (Ke et al., 2023) | RoBERTa (0.1B) | 🔥 | – | ✗ | ✗ |
| | ELLE (Qin et al., 2022) | BERT (0.1B) | 🔥 | memory+params | ✗ | ✗ |
| | CT0 (Scialom et al., 2022) | T0 (3B) | 🔥 | memory | ✗ | ✗ |
| | K-Adapter (Wang et al., 2021) | RoBERTa (0.3B) | ❄ | +params | ✗ | ✗ |
| | Gururangan et al. (2022) | GPT-2 (0.7B) | ❄ | +params | ✗ | ✗ |
| | CPT (Ke et al., 2022) | RoBERTa (0.1B) | ❄ | +params | ✗ | ✗ |
| | KILM (Xu et al., 2023a) | BART (0.4B) | 🔥 | – | ✗ | ✗ |
| | CaMeLS (Hu et al., 2023) | GPT-2 (1.5B) | 🔥 | auxiliary model | ✗ | ✗ |
| | SeMem (Peng et al., 2023b) | GPT-2 (0.7B) | ❄ | memory +auxiliary model | ✗ | ✗ |
| | CL-plugin (Lee et al., 2022a) | T5 (0.7B) | ❄ | +params | ✗ | ✗ |
| | Huang et al. (2023) | BERT (0.1B) | ❄ | +params | ✗ | ✗ |
| | GRACE (Hartvigsen et al., 2023) | T5 (0.06B) | ❄ | memory | ✗ | ✗ |
| **Memory -enhanced** | kNN-LM (Khandelwal et al., 2020) | ADP (Baevski and Auli, 2019) (0.2B) | ❄ | memory | ✔ | ✗ |
| | AdaptRet (He et al., 2021a) | ADP (0.2B) | ❄ | memory +auxiliary model | ✗ | ✗ |
| | RetoMaton (Alon et al., 2022) | ADP (0.2B) | ❄ | memory +auxiliary graph | ✗ | ✗ |
| | kNN-prompt (Shi et al., 2022) | GPT-2 (0.8B) | ❄ | memory | ✔ | ✔ |
| | Belief Bank (Kassner et al., 2021) | T5 (0.7B) | ❄ | memory +constraint solver | ✔ | ✔ |
| | FBNet (Tandon et al., 2022) | T5 (11B) | ❄ | memory +auxiliary model | ✗ | ✔ |
| | MemPrompt (Madaan et al., 2022) | GPT-3 (175B) | ❄ | memory+retriever | ✔ | ✔ |
| | TeachMe (Dalvi Mishra et al., 2022) | T5 (11B) | ❄ | memory+retriever | ✔ | ✔ |
| | SERAC (Mitchell et al., 2022b) | T5 (0.7B) | ❄ | memory +auxiliary model | ✗ | ✔ |
| | MeLLo (Zhong et al., 2023) | GPT-3.5 (175B) | ❄ | memory+retriever | ✔ | ✔ |
| **Retrieval -enhanced** | IC-Retrieval (Si et al., 2023) | GPT-3.5 (175B) | ❄ | retriever | ✔ | ✔ |
| | IC-RALM (Ram et al., 2023) | OPT (66B) | ❄ | retriever+reranker | ✗ | ✔ |
| | IKE (Zheng et al., 2023) | OPT (175B) | ❄ | retriever | ✔ | ✔ |
| | AAR (Yu et al., 2023b) | GPT-3.5 (175B) | ❄ | retriever | ✗ | ✔ |
| | RePlug (Shi et al., 2023b) | GPT-3 (175B) | ❄ | retriever | ✗/ ✔ | ✔ |
| | IRCoT (Trivedi et al., 2022) | GPT-3.5 (175B) | ❄ | retriever | ✔ | ✔ |
| | RARR (Gao et al., 2023) | PaLM (540B) | ❄ | search engine +auxiliary model | ✔ | ✔ |
| | RR (He et al., 2022) | GPT-3.5 (175B) | ❄ | retriever +auxiliary model | ✔ | ✔ |
| | ReFeed (Yu et al., 2023a) | GPT-3.5 (175B) | ❄ | retriever | ✔ | ✔ |
| | DecomP (Khot et al., 2023) | GPT-3.5 (175B) | ❄ | retriever | ✔ | ✔ |
| | ReAct (Yao et al., 2023a) | PaLM (540B) | ❄ | search engine | ✔ | ✔ |
| | Self-Ask (Press et al., 2023) | GPT-3 (175B) | ❄ | search engine | ✔ | ✔ |
| | FLARE (Jiang et al., 2023) | GPT-3.5 (175B) | ❄ | retriever/search engine | ✔ | ✔ |
| | DSP (Khattab et al., 2023) | GPT-3.5 (175B) | ❄ | retriever | ✔ | ✔ |
| | ART (Paranjape et al., 2023) | GPT-3.5 (175B) | ❄ | various tools | ✔ | ✔ |
| | Iter-RetGen (Shao et al., 2023) | GPT-3.5 (175B) | ❄ | retriever | ✔ | ✔ |
| | Verify-and-Edit (Zhao et al., 2023) | GPT-3.5 (175B) | ❄ | retriever/search engine | ✔ | ✔ |
| **Internet -enhanced** | Lazaridou et al. (2022) | Gopher (280B) | ❄ | search engine | ✔ | ✔ |
| | CRITIC (Gou et al., 2023) | GPT-3.5 (175B) | ❄ | various tools | ✔ | ✔ |
| | LLM Rewriter (Ma et al., 2023) | GPT-3.5 (175B) | ❄ | search engine | ✗ | ✔ |
| | Chameleon (Lu et al., 2023) | GPT-4 (?B) | ❄ | various tools | ✔ | ✔ |
| | ChatGPT Plugins (OpenAI, 2023a) | GPT-3.5 (175B) | ❄ | various tools | ✔ | ✔ |

Table 3: Comparison between representative methods. 🔥 means the parameters of the original LM are modified, while ❄ means they are unchanged; **Augmentation** means additional components used; **No Training** indicates the method does not require additional training; **Black-box** refers to whether the method suits non-publicly available models (*e.g.*, no model architecture, parameters, activations, or gradients are available). Note that we only list the largest size model used in the paper due to space limitations.

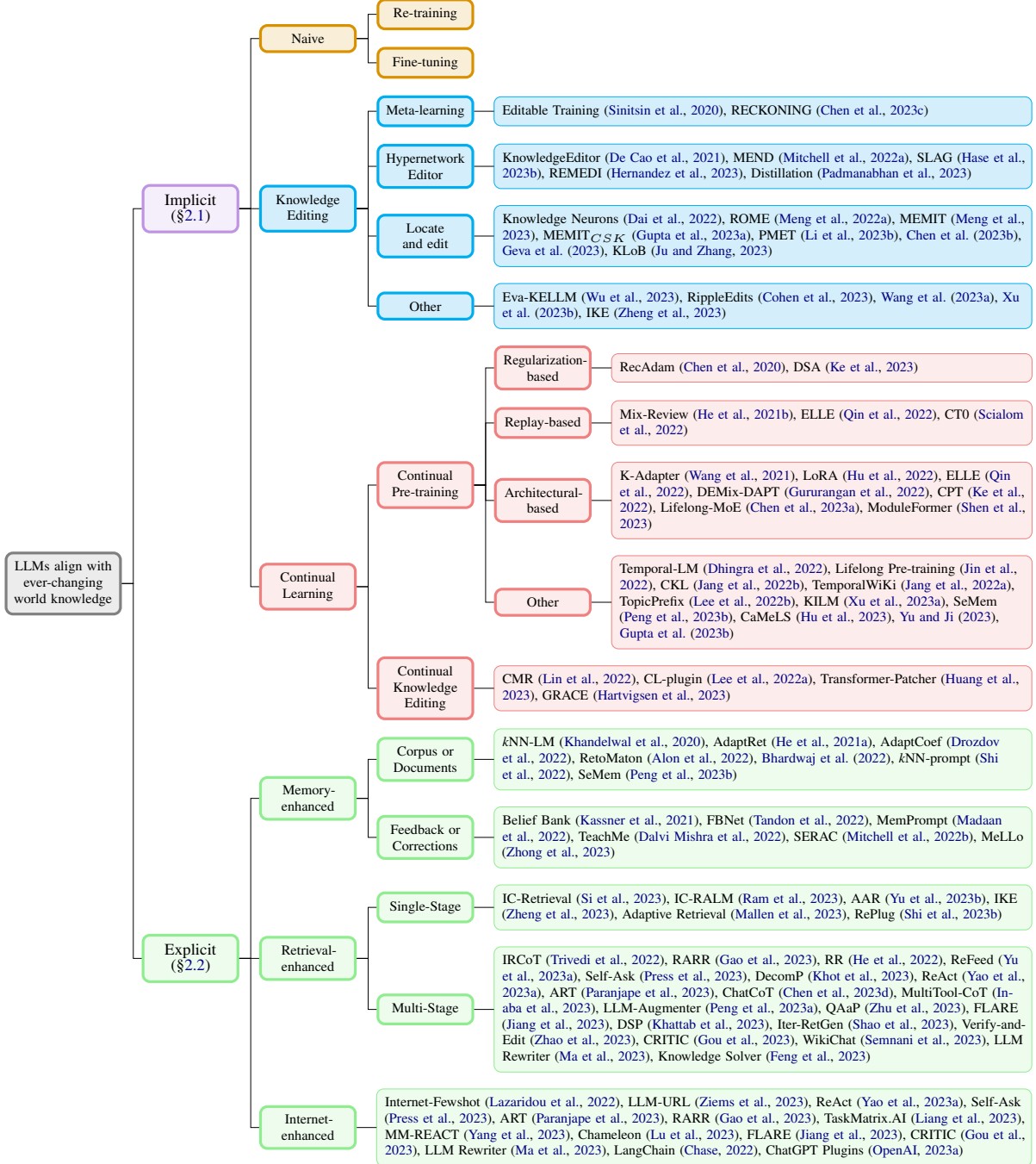

Figure 6: Taxonomy of methods to align LLMs with the ever-changing world knowledge. **Implicit** means the approaches seek to directly alter the knowledge stored in LLMs (*e.g.*, parameters) (§2.1), while **Explicit** means more often incorporating external resources to override internal knowledge (*e.g.*, search engine) (§2.2).