# OpenReview forum: "How Do Large Language Models Capture the Ever-changing World Knowledge? A Review of Recent Advances"
_EMNLP/2023/Conference — EMNLP 2023 Main_

### Official Review · Reviewer_CDPY · 2023-08-04

**Soundness:** 3

**Excitement:**

4: Strong: This paper deepens the understanding of some phenomenon or lowers the barriers to an existing research direction.

**Missing References:**

As mentioned above, Pan et al. (2023)

https://arxiv.org/pdf/2306.08302.pdf

Is a crucial source about research on updating LLMs with explicit knowledge.

**Paper Topic And Main Contributions:**

The paper analysis existing research on how large language models can take evolving world knowledge into account.

The authors create a taxonomy of approaches, with the broadest categorisation is "implicit" versus "explicit".

The authors restrict themselves to text-based knowledge.

**Reasons To Accept:**

The authors provide a useful overview for a sub area of research: keeping LLMs up to date with text-based knowledge sources.

**Reasons To Reject:**

The authors leave out a fast area of material and a key type of explicit knowledge. This can be called "keeping LLMs up to date with structural knowledge".

This type of assuring up-to-dateness is of high importance for LLM applications. And there are existing research reviews that show approaches to leverage structural data for improving LLMs, see esp. the paper from Pan et al. (2023)
https://arxiv.org/pdf/2306.08302.pdf

As the authors state, they cannot take into account all references. But looking at the references section of this proposed paper, and the reference section of Pan et al. (2023), one can see that the authors are not aware of a huge body of work related to explicit structural knowledge.

For this reason, I am deeply concerned about the paper in its current state. It would need a thorough re-write to take into account the work on structural knowledge for improving LLMs.

Such a rewrite could help a lot in closing the gap between research communities. But I am not sure if the authors can do this as part of an updated version of the paper, or rather would want to create a new paper on its own.

I have read the authors response from the rebuttal. Here are my responses.
- I agree that the timing for referencing Pan et al. in the initial submission does not work. Thank you for considering referencing the paper in the final submission!
- I agree that updating a KG is a challenge. However, I would say that Pan et al. and the combination of LLM+KG discussed in Pan et al., the part of "KG enhanced LLMs", is only about static knowledge.
- I disagree with the statement that "KG enhanced LLMs" is only about finetuning. In the rebuttal, the authors mention retrieval augmented generation as an example for updating an LLM with finetuning. As of today, there are many examples even in industry practice that demonstrate how to realise RAG with knowledge graphs. You can find one related blog post at https://neo4j.com/developer-blog/fine-tuning-retrieval-augmented-generation/ .
- A method which has more and more attention to implement RAG is the use of vector data bases. Again in industry practices, there is guidance on how this can be realised with KGs, and what the benefits of both approaches are, see https://neo4j.com/blog/knowledge-graph-vs-vectordb-for-retrieval-augmented-generation/.

I do not have a scientific reference at hand, but hope that these examples from industry are relevant for you.

**Reproducibility:**

N/A: Doesn't apply, since the paper does not include empirical results.

**Reviewer Confidence:**

4: Quite sure. I tried to check the important points carefully. It's unlikely, though conceivable, that I missed something that should affect my ratings.

---

> ### Author Rebuttal · Authors · 2023-08-29
>
> Dear Reviewer CDPY,
>
> We appreciate your valuable time, review and suggestions. We want to address your concerns in detail as follows.
>
> > Q1. Missing References: Pan et al. (2023)
>
> **A1:** Thanks for pointing out this related survey. However, Pan et al. (2023) was first released on 20 June on arXiv, three days before the EMNLP submission deadline and after our paper writing. It is regarded as **contemporaneous work** according to the policy. Having said that, we will cite this paper and compare it when we can modify our paper.
>
> > Q2. The authors leave out a fast area of material and a key type of explicit knowledge. This can be called "keeping LLMs up to date with structural knowledge".
>
> **A2:**
> Thanks for your comments. We acknowledge that structural knowledge such as knowledge graphs (KGs) can broadly be categorised as explicit knowledge, which can be retrieved from external to adapt LLMs to new world knowledge.
>
> However, please note that we consider the issue that LLMs can be outdated after **deployment/training**, and how to align trained LLMs to the **ever-changing** world knowledge. We focus on **dynamic**, not static, as stated in multiple places in our paper, e.g. L007, L332, L446, L1616, L1622.
>
> After creation, KG is static and can still be outdated [Ji et al. (2021)]. New information or discoveries not yet incorporated into the graph may lead to potentially outdated knowledge. *Therefore, simply incorporating structural knowledge (i.e., KG) into LLM does not solve our problem.*
>
> Unfortunately, mainstream research focuses on static KGs [Ji et al. (2021), Page 17, Section F], and there is little work on using KGs to update new knowledge for LLMs to reflect the most recent or unseen facts and knowledge. Having said that, if any applicable approaches are missing, we are delighted to add them.
>
> On the other hand,  our survey scope is to align already trained/deployed LLMs (such as Llama, GPT4) with the ever-changing world knowledge **without retraining from scratch**. From our best understanding, most of the existing KG-enhanced LLM approaches require special pre-training objectives to integrate KGs into LLMs, which need to be retrained from scratch and can **not** be applied to a deployed LLM [Pan et al. (2023), Section 4.1]. For example, as summarized in Pan et al. (2023) Table 2, both REALM[1] and RAG[2] require joint training of the retriever and the model (we have discussed REALM and RAG in our paper as well at L332).
>
> "Keeping LLMs up to date with structural knowledge" is a good idea and could be another paper. We will add discussion in the revision. Thanks.
>
> Ji et al. (2021): A Survey on Knowledge Graphs: Representation, Acquisition and Applications, 2021\
> Pan et al. (2023): Unifying Large Language Models and Knowledge Graphs: A Roadmap, 2023\
> [1] REALM: Retrieval-Augmented Language Model Pre-Training, 2020\
> [2] RAG: Retrieval-Augmented Generation for Knowledge-Intensive NLP Tasks, 2020
>
> > Q3. This type of assuring up-to-dateness is of high importance for LLM applications. And there are existing research reviews that show approaches to leverage structural data for improving LLMs, see esp. the paper from Pan et al. (2023)
>
> **A3:**
> We agree that structural data is useful for improving LLMs and has a lot of potential. However, for the broader structural knowledge, including KGs and other types, there is **rare** work on incorporating structural knowledge such as KG into LLMs for updating data and **without retraining**. In Pan et al. (2023), most works for improving LLMs are about overcoming hallucinations, improving interpretability, and accessing domain-specific knowledge. (We just mentioned KG-enhanced LLMs here. Other parts in their paper, such as LLM-augmented KGs and Synergized LLMs + KGs, are not relevant.). The current *“leverage structural data for improving LLMs”* does not fully align with our scope. In addition, we have compared our work with related knowledge-enhanced LM surveys in the Appendix (L1616).
>
> We will add future work to discuss it. If the KGs can be easily updated with recent and new knowledge, it can provide a way to work with LLMs for new knowledge. Again, if any applicable approaches are missing, we would be delighted to add them.
>
> > Q4. As the authors state, they cannot take into account all references. But looking at the references section of this proposed paper, and the reference section of Pan et al. (2023), one can see that the authors are not aware of a huge body of work related to explicit structural knowledge.
>
> **A4:**
> There are two reasons for having different reference sections with Pan et al. (2023).
>
> **First, our survey scope is different from Pan et al. (2023).**
> There may be a misunderstanding about our scope. The scope of Pan et al. (2023) is about *KG-enhanced LLMs*, *LLM-augmented KGs*, and *Synergized LLMs + KGs*. Only *KG-enhanced LLMs* is relevant to us because we aim to improve LLMs instead of improving KGs. In the KG-enhanced LLMs, they discuss hallucinations, interpretability, and accessing domain-specific knowledge. They do not mention ever-changing knowledge. As mentioned in the abstract and in the above answers, our scope targets the **ever-changing** world knowledge **without re-training**.
>
> **Second, our surveyed time range differs from Pan et al. (2023).**
> As mentioned in the Limitations Section (L647), we review papers from the **three** recent years (2021-2023), mostly in 2022 and 2023. In contrast, Pan et al. (2023) mainly review papers before 2022. For KG-enhanced LLM methods, as shown in Table 2 in their paper, most of the methods are before 2021, not to mention these methods do not apply to trained models.
>
> Due to the above reasons, our references section does not entirely overlap with Pan et al. (2023), but this does not mean we are unaware of the work related to explicit structural knowledge.

---

### Official Review · Reviewer_PyMv · 2023-08-04

**Soundness:** 4

**Excitement:**

4: Strong: This paper deepens the understanding of some phenomenon or lowers the barriers to an existing research direction.

**Paper Topic And Main Contributions:**

This paper presents an extensive and detailed compendium of the different methods developed in recent years to align LLMs with updated world knowledge. The paper proposes a taxanomy to structure this study where the different methods are divided in two main faimilies: implicit, where the approaches seek to directly alter the internal representations of the LLM, and explicit, where external resources are incorporated to override internal knowledge during inference.

In addition to the exhaustiveness of this work, I would also highlight how the authors put in context each of the different approches covered in the study, providing a perspective of what methods may make more sense in what situations, and offering a perspective of future trends and challenges.

The only criticism I have is that I would have liked to see a quantitative study that allowed to understand the effectiveness of the different methods. This could have been done, e.g. by selecting those LLMs that have been updated in past work through different approaches and showing the relaitve impact of each such approach on the same LLM.

**Reasons To Accept:**

+ Exhaustive, complete, and very useful
+ Provides a notion of what methods may make more sense in what situations
+ Includes a perspective of future trends and challenges.

**Reasons To Reject:**

- I would have liked to see a quantitative studyin the paper, too.

**Reproducibility:**

N/A: Doesn't apply, since the paper does not include empirical results.

**Reviewer Confidence:**

4: Quite sure. I tried to check the important points carefully. It's unlikely, though conceivable, that I missed something that should affect my ratings.

---

> ### Author Rebuttal · Authors · 2023-08-29
>
> Dear Reviewer PyMv,
>
> We appreciate your valuable time, review and suggestions. We want to address your concerns in detail as follows.
>
> > Q1. I would have liked to see a quantitative studyin the paper, too.
>
> **A1:**
> As discussed in the Limitations Section (L671), we acknowledge that we do not have quantitative comparisons of different approaches through experiments but only empirical analysis and summarization of existing work. We also agree that implementing different approaches on the same LLM would make comparing various methods easier and more straightforward.
>
> However, this **unified** evaluation of different categories is **not trivial** -- existing work evaluates their methods in the same category, and there are no unified suitable benchmarks, datasets, and settings for fair comparison of different categories. For example, knowledge editing methods typically evaluate the fact-checking tasks to verify that knowledge has been updated, while continual pre-training usually evaluates both the source and target domains. It might be unfair to say which category is more effective without a unified evaluation setting.
>
> That's why we provide a qualitative analysis of different categories in this survey to help choose suitable approaches under different scenarios.  In fact, as we discussed in Section 4 (L607), quantitative evaluation (benchmarks) is a challenging and interesting future direction to fairly compare methods of different categories to align LLMs with updated world knowledge. We will leave quantitative comparisons and analysis as future work.

---

### Official Review · Reviewer_j2Rw · 2023-08-11

**Soundness:** 3

**Excitement:**

4: Strong: This paper deepens the understanding of some phenomenon or lowers the barriers to an existing research direction.

**Paper Topic And Main Contributions:**

The paper surveys techniques related to LLM alignment over changing information that does not require training from scratch. The paper is comprehensive in nature.

**Reasons To Accept:**

Solid and comprehensive work.

**Reasons To Reject:**

Not clear how the papers are selected (i.e., survey methodology) and the search timeframe (last N years?).



**Reproducibility:**

N/A: Doesn't apply, since the paper does not include empirical results.

**Reviewer Confidence:**

1: Not my area, or paper was hard for me to understand. My evaluation is just an educated guess.

---

> ### Author Rebuttal · Authors · 2023-08-29
>
> Dear Reviewer j2Rw,
>
> We appreciate your valuable time, review and suggestions. We want to address your concerns in detail as follows.
>
> > Q1. Not clear how the papers are selected (i.e., survey methodology)
>
> **A1:** Due to the limited space in the paper's main body, we have briefly described how we selected papers in the Limitations Section (L642). Specifically, we mainly review cutting-edge papers from the top conferences, including ACL, EMNLP, NAACL, TACL, NeurIPS, ICML, ICLR, and some of the most recent papers from arXiv. We find papers from these venues via keyword searching from the paper title and abstract. Then, we manually filter out irrelevant papers and categorize the remaining into different approaches.
>
> > Q2. and the search timeframe (last N years?).
>
> **A2:**
> As mentioned in the Limitations Section (L647), we review papers from the **three** recent years (2021-2023), mostly in 2022 and 2023. We set the timeframe as three years because:
> 1. the concept of “large language models” arose within three years
> 2. much work has been proposed in aligning large language models with world knowledge, especially after the release of the ChatGPT
>
> We want to review the recent advances in the LLMs era.

---

### Meta-Review · Area_Chair_8oCk · 2023-09-10

**Recommendation:** 5

**Metareview:**

The main conclusions of the reviews and the post-rebuttal discussions:
- 3/ 3 reviewers consider the paper sound (scores 3, 4, 3)
- 3/ 3 reviewers find the paper exciting (scores 4, 4, 4)

Front reading the rebuttal and seeing the scores above, I find that the reviewers consider strong points for soundness the following:
- the paper provides a comprehensive survey on LLM alignment over changing information that does not require training from scratch
One concern is no quantitative comparisons of different approaches through experiments but only empirical analysis and summarization of existing work and it is addressed in the limitations section of the paper.
Concerns from R3 about inclusion of more citations is clarified in the rebuttal.

All reviewers find the paper very exciting!

---

### Decision · Program_Chairs · 2023-10-07

**Decision:**

Accept-Main

**Comment:**

The main conclusions of the reviews and the post-rebuttal discussions:
- 3/ 3 reviewers consider the paper sound (scores 3, 4, 3)
- 3/ 3 reviewers find the paper exciting (scores 4, 4, 4)

Front reading the rebuttal and seeing the scores above, I find that the reviewers consider strong points for soundness the following:
- the paper provides a comprehensive survey on LLM alignment over changing information that does not require training from scratch
One concern is no quantitative comparisons of different approaches through experiments but only empirical analysis and summarization of existing work and it is addressed in the limitations section of the paper.
Concerns from R3 about inclusion of more citations is clarified in the rebuttal.

All reviewers find the paper very exciting!